# A Re-Appraisal of the Effect of Amplitude on the Stability of Interlimb Coordination Based on Tightened Normalization Procedures

**DOI:** 10.3390/brainsci10100724

**Published:** 2020-10-13

**Authors:** Harjo J. de Poel, Melvyn Roerdink, C. (Lieke) E. Peper, Peter J. Beek

**Affiliations:** 1Department of Human Movement Sciences, University Medical Center Groningen (UMCG), University of Groningen, Antonius Deusinglaan 1, 9713 AV Groningen, The Netherlands; 2Department of Human Movement Sciences, Faculty of Behavioural and Movement Sciences, Vrije Universiteit Amsterdam, Amsterdam Movement Sciences, van der Boechorststraat 7-9, 1081 BT Amsterdam, The Netherlands; m.roerdink@vu.nl (M.R.); c.e.peper@vu.nl (C.E.P.); p.j.beek@vu.nl (P.J.B.)

**Keywords:** rhythmic movement, synchronization, entrainment, bimanual interaction, dynamical systems, coordination dynamics, relative phase, perturbations

## Abstract

The stability of rhythmic interlimb coordination is governed by the coupling between limb movements. While it is amply documented how coordinative performance depends on movement frequency, theoretical considerations and recent empirical findings suggest that interlimb coupling (and hence coordinative stability) is actually mediated more by movement amplitude. Here, we present the results of a reanalysis of the data of Post, Peper, and Beek (2000), which were collected in an experiment aimed at teasing apart the effects of frequency and amplitude on coordinative stability of both steady-state and perturbed in-phase and antiphase interlimb coordination. The dataset in question was selected because we found indications that the according results were prone to artifacts, which may have obscured the potential effects of amplitude on the post-perturbation stability of interlimb coordination. We therefore redid the same analysis based on movement signals that were normalized each half-cycle for variations in oscillation center and movement frequency. With this refined analysis we found that (1) stability of both steady-state and perturbed coordination indeed seemed to depend more on amplitude than on movement frequency per se, and that (2) whereas steady-state antiphase coordination became less stable with increasing frequency for prescribed amplitudes, in-phase coordination became more stable at higher frequencies. Such effects may have been obscured in previous studies due to (1) unnoticed changes in performed amplitudes, and/or (2) artifacts related to inappropriate data normalization. The results of the present reanalysis therefore give cause for reconsidering the relation between the frequency, amplitude, and stability of interlimb coordination.

## 1. Introduction

When one moves multiple limbs simultaneously, the limb movements are not independent but influence each other. As a result, some rhythmical interlimb tasks are relatively easy to perform, such as cycling both hands at the same frequency in either symmetrical (in-phase) or anti-symmetrical (antiphase) fashion, whereas other rhythmic interlimb tasks are quite difficult to perform without training, such as cycling both hands with a quarter cycle difference between them or at two different frequencies [1]. From the perspective of dynamical systems theory, or coordination dynamics in short, such phenomena are understood in terms of pattern stability, with some patterns being intrinsically stable and others being intrinsically unstable. Rhythmical interlimb behavior is thereby characterized by attraction to a limited number of stable coordination patterns [1,2]. Stable interlimb coordination occurs by virtue of the coupling between the components involved: When there is no interaction between limb movements there can be no attraction towards synchronized/coordinated patterns (e.g., [2,3]). Empirically, this is evidenced by the existence of certain stable and hence preferred modes of interlimb coordination in animals and humans [1,4]. A seminal finding in this regard is the well-known observation that the stability of a coordination pattern may be lost when movement frequency is increased [5,6]. Such destabilizations may lead to a spontaneous switch in coordination. For rhythmic interlimb coordination this entails an unintended switch from the less stable antiphase pattern (movements of the two limbs are exactly alternating: Relative phase ≈ 180°) to the more stable in-phase pattern (movements of the two limbs are exactly coinciding: Relative phase ≈ 0°). This phenomenon was formalized in the influential Haken–Kelso–Bunz (HKB) model [2], which consists of a potential function (or gradient dynamics) for relative phase and a system of coupled oscillators underlying the gradient dynamics of relative phase. An entire research program for studying interlimb (e.g., [1,7,8,9]), sensorimotor (e.g., [1,10,11,12,13]), and between-person (e.g., [1,3,14,15]) coordination has been built on the HKB model, which has become a cornerstone of coordination dynamics.

In the coupled-oscillator model in question [2], the strength of the interaction between the two involved oscillators and, hence, the dynamics of the coordination patterns (e.g., stability) are either completely or partly mediated by frequency-induced changes in amplitude rather than by the frequency as such [2,16]. However, while ample research has been conducted in this context that focused on the effects of manipulations of movement frequency, the role of movement amplitude has remained underexposed. This is unfortunate because it is generally known that higher frequencies are accompanied by smaller characteristic amplitudes and vice versa that larger amplitudes are accompanied by lower frequencies [17,18]. Theoretically, therefore, the typically reported effects of movement frequency on coordinative stability (e.g., [1,9,15]) could thus also be engendered by frequency-induced changes in the amplitudes of the limb movements. For the vast majority of previous studies this cannot be verified because the performed amplitudes were neither reported nor considered in interpreting the results, which may have introduced potentially unjustified and confounded conclusions regarding coordinative stability as a function of movement frequency. Previous studies have indeed found indications that moving with larger amplitudes relates to enhanced interlimb [19,20,21,22,23,24,25,26] and perceptuo-motor coupling [27,28,29]. With the current study, we opt for a re-appraisal of the role of amplitude on coordinative stability (as was done for the first time two decades ago by, e.g., [16,19,20]) by presenting results of a refined, artifact-free analysis of the experimental data of Post, Peper, and Beek [21].

### 1.1. Incentives for a Refined Analysis of Post, Peper, and Beek (2000) 

We selected the data of Post et al. [21] for the following reasons. First of all, the study was designed specifically to tease apart the effects of amplitude and frequency on coordinative stability in order to test the predictions of the HKB coupled-oscillator dynamics [2]. In the experiments in question, six participants performed in-phase and antiphase movements around the elbow joints at seven different movement frequencies (0.75 Hz to 2.25 Hz) at three prescribed amplitude conditions (0.1, 0.2, and 0.3 rad) and a ‘free’ amplitude condition. Furthermore, and of great relevance for the present purposes, the experimental design involved a controlled mechanical arrest of one of the moving limbs. Arresting the limb for a quarter of a cycle, as was done in the experiment (see the Methods section for details), resulted in a perturbation of the interlimb coordination pattern and a subsequent return to the original coordination pattern. From this return or relaxation trajectory, direct estimates of pattern stability can be derived empirically, in addition to the more commonly used ‘indirect’ steady-state measures of pattern stability, such as the standard deviation of relative phase [30]. Results in the original paper by Post et al. [21] showed that both in- and antiphase coordination patterns were more variable for smaller prescribed amplitudes (i.e., lower steady-state coordinative stability), but this was not corroborated by slower responses to relative-phase perturbations (i.e., which would have signaled a lower stability for smaller amplitudes). 

Counter to these outcomes, however, the results of more recent perturbation studies with the same apparatus and similar but slightly different experimental designs and outcome measures, showed that the interlimb coupling strength scaled quite substantially with movement amplitude [22,23]. It was deemed possible, therefore, that in the original study of Post et al. [21], the perturbation outcomes were prone to artifacts arising from the data analysis, which could have invalidated the empirical evaluation of the effect of amplitude on coordinative stability. In particular, the exemplary post-perturbation relative-phase data reported by Post et al. (see Figure 1a,b in [21]) fueled this concern and prompted us to reanalyze those data using tightened normalization procedures, as will be delineated next.

### 1.2. Normalization Issues

Post et al. [21] determined the phase angle (θ) of the individual limb movements (indicating the instantaneous phase in their respective cycles) to estimate the continuous relative phase φ between the limb movements (i.e., by calculating the phase difference: $\varphi =\theta_{1}-\theta_{2}$). From this continuous estimate of relative phase steady-state variability and relaxation characteristics were extracted as indices of coordinative stability and coupling. It is important to consider, however, that, depending on the specific research question of interest and the characteristics of the movement signals at hand, the calculation of θ may require several normalization steps to avoid artifacts in the evolution of θ and thus φ. Since a correct phase determination requires near-to-harmonic oscillations [31,32], several methodological studies have highlighted the importance of appropriate normalization of the time series prior to calculating θ to prevent computational artifacts due to (1) (variations in) oscillation frequency, and (2) offset of the oscillation center [33,34] (see [35] for an overview). Other sources of artifacts that have received less attention are gradual and/or sudden changes in amplitude and/or the center of oscillation over cycles [22,36]. By analyzing generated signals with known properties, De Poel [36] demonstrated how such changes may introduce substantial artifacts in θ (see Appendix B) and thus φ, which may result in invalid (or even antithetical) interpretations. In fact, such artifacts may readily occur when rather capricious and/or non-stationary/non-steady-state ‘cyclical’ movements are made, as in experiments involving pattern transitions and transient mechanical perturbations, as in the study of Post et al. [21]. In particular, applying a transient mechanical perturbation of the arm movement perturbs all aspects of the movement, including the amplitude and center of oscillation. This is exemplified in Figure 1a: Directly after the perturbation and around 1 s, the right (perturbed) arm’s movement (red line) shifts away from the average positioning of zero (in this case implying that the limb moved somewhat closer towards the body) before regaining position approximately 4 cycles later.

In this regard, Post et al. [21] reported observing undulations in φ directly after perturbations, as exemplified in Figure 1e. While inspecting the analogous original figures in [21] (see also [37,38]) we realized that the frequency of the undulations in φ closely matched the actual movement frequency of the limbs, which suggested that potential artifacts had occurred. As we now know based on signals with known properties [36] (see Appendix B), these undulations were indeed likely due to within-trial changes in the position of the center of oscillation (which were readily observed, see the exemplary data in Figure 1a). The varying oscillation center implies that the corresponding phase portrait is biased away from the origin (Figure 1c), hence introducing artifacts in the form of undulations in θ (in the example especially for the right arm, see Figure 1d) and thus φ (Figure 1e).

Potential solutions to prevent such artifacts include detrending the oscillation center, for instance by high-pass filtering (see, e.g., [39,40]), or normalizing the oscillation center each half-cycle (e.g., [22,36,41], see also Appendix B). Since the latter method is to be preferred because it also corrects for all previously mentioned causes of artifacts, we applied this method to the data of Post et al. [21]. (Note that in studies subsequent to Post et al. [21,37] with a similar perturbation procedure, such normalization per half cycle was already implemented [22,23]). Based on the above, we expected that the reduced undulations in the post-perturbation φ-signal would lead to more valid (i.e., artifact-free) and reliable (i.e., better fitting) estimations of the perturbation-related stability parameter. We suspected that a reanalysis of the data of Post et al. [21] in this manner would yield more prominent effects of amplitude on pattern stability (as derived from perturbation analysis) than those previously reported. To anticipate, the reanalysis indeed showed that interlimb pattern stability depended more on amplitude than on movement frequency per se. It further revealed that steady-state antiphase coordination with prescribed movement amplitudes became less stable with increasing movement frequencies, whereas in-phase coordination became more stable. Although there were only six participants in the study of Post et al. [21], implying that the present findings should be interpreted with caution, they provide strong initial incentives for rekindling empirical research on the role of amplitude in coordinative stability.

## 2. Materials and Methods

We reanalyzed the data reported in Post et al. in the manner described in that article [21], except for the normalization prior to phase determination. For specific details we refer to [21]. Here we briefly reiterate and highlight the most important aspects.

### 2.1. Participants

Pre-existing data of 6 healthy female right-handed volunteers (20–27 years) were used. At the time of the experiment, these 6 participants were selected from a larger pool solely based on their ability to perform in-phase and antiphase movements in the experimental amplitude and frequency ranges as determined from a preceding test session (see [21]). Participants engaged in the experiment after having provided informed consent and the experiment was conducted in accordance with the Declaration of Helsinki.

### 2.2. Apparatus

Participants were seated in a modified chair with mobile armrests that allowed for rotation around the elbow joint in the horizontal plane. The angular displacement of the armrests was captured with a hybrid potentiometer (22HHPS-10 Sakae) at a sampling rate of 300 Hz. A digital actuator controller (developed by Fokker Aerospace) controlled a pair of torque motors mounted below the axis of the armrests. These motors could block one of the armrests based on current positions and velocities and in this way controlled perturbations of the coordination patterns could be applied. Amplitudes were prescribed using LED indicators for maximal excursions of each arm, and movement frequencies were prescribed by means of a digital metronome (one beep per full movement cycle). 

### 2.3. Procedure

In the experiment, three factors were manipulated: Coordination pattern (in-phase and antiphase), movement frequency (one unpaced and seven paced conditions, ranging from 0.75 to 2.25 in steps of 0.25 Hz), and movement amplitude (one ‘free’ amplitude condition and three prescribed amplitude conditions of 0.1, 0.2, and 0.3 rad; note that for clarity, in this paper the angular displacement of the arms is expressed in radians, while (relative) phase angles are expressed in degrees. Amplitude was defined as half the peak-to-peak range. Each participant performed all combinations 6 times and in 5 of these trials the right arm was perturbed, while one trial was a catch trial without perturbation. The in-phase and antiphase trials were presented in blocks (in randomized order), and within these blocks the prescribed amplitude was also presented in blocks (also in randomized order). The free amplitude trials were presented in a separate block. The order of presentation was fully randomized within each amplitude block. As the main focus of the reanalysis was on effects of amplitude and frequency, we did not consider the trials with a freely chosen frequency.

The mechanical perturbation consisted of a full arrest of the right arm close to the moment of elbow extension (where velocity is near zero). The arrest lasted for a quarter of a movement cycle (as calculated from the preceding three cycles), after which the arm was released again. In other words, the interlimb pattern (of around either 0° or 180° relative phase) was perturbed by 90°. The perturbation was applied at random between the 12th and 17th cycle of the trial.

### 2.4. Data Reduction 

Angular-displacement time series were differentiated, and subsequently both angular displacement and velocity were low-pass filtered using a bidirectional 2nd order Butterworth filter (cut-off frequency 25 Hz). Peak angular excursions were determined with a custom-made peak-picking algorithm. These peak indices were used as indicators of the beginning and end of each half cycle. Rather than normalizing the time series by an average over the whole trial (as done in Post et al. [21]), we now normalized for each half cycle separately [36] (see also [22]). Specifically, within each half-cycle bin, the angular displacement of each arm was normalized for the angular frequency [31] of the respective half cycle [34]. Most importantly for the present purposes, the position signal was additionally centered per half cycle by subtracting the center value (i.e., midway between minimum and maximum) of the respective half-cycle bin. In this way, we attained a ‘closer-to-circle-shaped’ phase portrait (i.e., the velocity plotted against the position signal, Figure 1b,c), which was suitably centered at the origin for the entire trial time series. Phase angles ($\theta_{L}$ and $\theta_{R}$, for the left and right arm, respectively; see Figure 1d) could subsequently be determined by taking the angle of the polar coordinates in phase space. The continuous relative phase (see Figure 1e) was determined as $\varphi =\theta_{R}-\theta_{L}$. Figure 1d,e demonstrate how this ‘half-cycle centering’ refined the relative-phase signals. For further analysis, the φ-signal was segmented into a pre-perturbation steady-state segment (8 cycles prior to perturbation onset), a perturbation transient segment (3 s following perturbation release: $\varphi_{\mathrm{return}}$), and a post-perturbation steady-state segment (8 cycles following the return segment).

### 2.5. Analysis

Steady-state performance was evaluated in terms of performed movement frequency and amplitude (both determined using the peak indices), and the variability of the relative phase (standard deviation according to statistical methods for directional data [42]) over pre-perturbation ($SD\varphi_{pre}$) and post-perturbation segments ($SD\varphi_{post}$). In the remainder of this paper, the values of $SD\varphi_{pre}$ are used to indicate steady-state coordinative stability (SDφ).

Coordinative stability was further quantified by the strength of the return process of φ after the 90° perturbation. This could be determined by fitting the return signal (starting from a value of 45° from the mean value of $\varphi_{pre}$) to an exponential decay function (including a damped oscillation element, see [21]): (1)$$\varphi \left(t\right)=p+qe^{-\lambda t}\cos\left(\omega_{\varphi }t+\theta \right),$$
in which the decay parameter λ indicates the speed of the return process, also known as the relaxation rate. As can be appreciated from Equation (1), higher values of λ indicate faster relaxation to the according coordination pattern, and thus higher stability. The estimate of the decay parameter λ can therefore be used as an index of stability. An example of the outcome of the fit procedure is provided in Figure 1. For more specific details about this process, we refer to [21].

A trial was excluded from further analysis if one or more of the following criteria were met (see also [21]): (a) The original coordination pattern was not re-established after the perturbation; (b) the entire return signal remained > 45° from the pre-perturbation steady-state pattern; (c) pre-perturbation and/or post-perturbation performance was not stable ($SD\varphi_{pre}$ and/or $SD\varphi_{post}$ > 45°); (d) the decay fit showed no decay (i.e., λ<0); and (e) the fit was unreliable as indicated by lack of robustness over four different initial values of the fit model (standard error(λ)>median(λ)). Accordingly, 157 out of 1908 trials (i.e., 8%) were excluded from further analysis (note that this was 21% in [21]).

### 2.6. Statistical Analysis

The stability indices SDφ and λ were analyzed statistically using Repeated-Measures (RM) ANOVAs with the within-subjects factors pattern, frequency, and (where applicable) amplitude. Estimates of effect sizes were provided by means of the ‘partial eta squared’ ($\eta_{p}^{2}$). Obtained main and interaction effects were scrutinized further based on the analysis of the pertinent simple effects and (if applicable) subsequent post-hoc pair-wise comparisons based on *t*-tests with Holm–Bonferroni correction for multiple comparisons. Further details on the post-hoc treatment for each test are provided in the Results section.

## 3. Results

Figure 2 summarizes the frequency-amplitude relations. As already reported in Post et al. [21], participants on average closely adhered to the prescribed amplitudes, while for the free amplitude condition the performed amplitudes generally dropped with increasing frequencies. Regarding the latter, a 2 pattern × 7 frequency RM ANOVA on the performed ‘free’ amplitudes yielded a significant main effect of frequency, *F*(6,30)=18.96, *p* < 0.001, $\eta_{p}^{2}$ = 0.79, and a significant pattern × frequency interaction, *F*(6,30) = 3.44, *p* = 0.01, $\eta_{p}^{2}$ = 0.41. Post-hoc analysis of this interaction effect revealed that the simple effect of frequency was significant for both in-phase and antiphase. Subsequent pair-wise comparisons for in-phase indicated significantly larger performed amplitudes for 0.75 Hz compared to all other frequencies but 1 Hz (all *p*s < 0.01), for 1 Hz compared to the highest three frequencies (1.75–2.25 Hz; all *p*s < 0.01), and for 1.25 Hz compared to 2.25 Hz (*p* < 0.05). For antiphase the performed amplitude differed statistically for 0.75 Hz compared to all other frequencies (all *p*s < 0.001), and between pairs 1-2 Hz (*p* = 0.005), 1–2.25 Hz (*p* < 0.001), and 1.25–2.25 Hz (*p* = 0.03). For both in-phase and antiphase the performed amplitudes did not differ statistically between the highest four frequencies (1.5–2.25 Hz; all *p*s > 0.05). Notably, for 0.75 and 1 Hz the performed amplitude in the free amplitude condition was larger than that of the largest prescribed amplitude of 0.3 rad (all *p*s < 0.01). For outcomes of individual participants, see the Appendix A.

The statistical analysis of SDφ for the prescribed amplitude conditions involved a 2 pattern × 7 frequency × 3 amplitude RM ANOVA. Antiphase coordination (mean SDφ = 9.7°) was significantly more variable than in-phase coordination (mean SDφ = 6.9°) as indicated by a significant main effect for pattern, *F*(1,5) = 172.08, *p* < 0.001, $\eta_{p}^{2}$ = 0.97. For amplitude, a significant main effect was observed, *F*(1,10) = 8.64, *p* = 0.007, $\eta_{p}^{2}$ = 0.63. With smaller amplitude, SDφ generally increased (see also Figure 3a,b), indicating lower coordinative stability. Post-hoc analysis indicated that SDφ at 0.1 rad (mean SDφ = 9.3°) differed significantly from the other two conditions (0.2 rad: 8.0°, 0.3 rad: 7.7°). Note that Post et al. [21] also found this effect. 

Regarding the effects of frequency an unanticipated result was observed. A significant main effect for frequency was found, *F*(6,30) = 8.73, *p* < 0.001, $\eta_{p}^{2}$ = 0.64, indicating that, collapsed over in-phase and antiphase patterns, SDφ increased. Remarkably, however, this commonly observed finding was accompanied by a significant pattern × frequency interaction, *F*(6,30) = 24.22, *p* < 0.001, $\eta_{p}^{2}$ = 0.83, with simple effects for pattern showing the expected significant increase in SDφ with increasing movement frequency for antiphase coordination (mean SDφ: 8.6°, 8.5°, 8.0°, 8.8°, 10.0°, 11.3°, and 11.9°, for 0.75–2.25 Hz respectively; see also Figure 3b), but a significant decrease in SDφ for in-phase coordination (mean SDφ: 8.7°, 7.8°, 6.9°, 6.5°, 6.4°, 6.5°, and 6.2°, for 0.75–2.25 Hz respectively; see also Figure 3a). According to the post-hoc pair-wise comparisons of these simple frequency effects, in-phase performance at the lowest two frequencies (i.e., 0.75 and 1 Hz) was statistically more variable than at all subsequent frequencies (*p*-values < 0.01), while antiphase performance at the highest three frequencies (i.e., 1.75, 2, and 2.25 Hz) was statistically more variable than at the lower frequencies (all *p*-values < 0.01), except for the one directly preceding it (e.g., the 1.5–1.75 Hz comparison was not significant). Individual participant outcomes are provided in the Appendix A. 

For the prescribed amplitude conditions, a 2 pattern × 7 frequency × 3 amplitude RM ANOVA on λ also revealed a significant main effect of amplitude, *F*(1,10) = 7.01, *p* = 0.01, $\eta_{p}^{2}$ = 0.58, which indicated that stability increased with amplitude: Overall mean values of λ were 2.05 (0.1 rad), 2.24 (0.2 rad), and 2.43 (0.3 rad). Note that Post et al. [21] did not find this significant effect of amplitude. Furthermore, a significant main effect of frequency was obtained, *F*(6,30) = 4.45, *p* = 0.002, $\eta_{p}^{2}$ = 0.47, accompanied by a significant pattern × frequency × amplitude interaction, *F*(12,60) = 2.86, *p* = 0.004, $\eta_{p}^{2}$ = 0.36. Based on post-hoc analysis of simple effects of frequency for each pattern × amplitude combination and subsequent pair-wise comparisons, the three-way interaction appeared to be primarily due though to the rather sharp increase of λ in the fastest condition (2.25 Hz) but only for the largest two amplitudes in antiphase coordination (see Figure 4b). Notably, there was no significant main effect of pattern, as was the case in [21].

For the free amplitude conditions, the 2 pattern × 7 frequency RM ANOVA on SDφ revealed a significant main effect of pattern: Antiphase coordination (mean SDφ = 9.2°) was more variable than in-phase coordination (mean SDφ = 6.3°), *F*(1,5) = 78.48, *p* < 0.001, $\eta_{p}^{2}$ = 0.94. Furthermore, a significant main effect of frequency, *F*(6,30) = 4.49, *p* = 0.002, $\eta_{p}^{2}$ = 0.48, accompanied by a significant pattern × frequency interaction, *F*(6,30) = 2.66, *p* < 0.03, $\eta_{p}^{2}$ = 0.35, and subsequent analysis of the simple main effects for in-phase and antiphase separately, revealed that for in-phase the frequency effect did not reach significance (*p* = 0.08; SDφ remained rather similar for all frequencies, see Figure 3a, closed circles), while for antiphase SDφ increased significantly (*p* < 0.001) with frequency (Figure 3b, closed circles), in which post-hoc pair-wise comparisons only indicated that performance at 2.25 Hz differed significantly from that at 0.75 and 1 Hz (*p*-values < 0.05).

For the condition with unprescribed (‘free’) amplitude, a 2 pattern × 7 frequency RM ANOVA on the values of λ yielded no significant effects. For λ-outcomes of individual participants, see the Appendix A.

## 4. Discussion

We reanalyzed the interlimb coordination data of Post et al. [21] with the expectation that the refined analysis (based on improved normalization procedures, see Appendix B) would yield clearer effects of amplitude on the stability of steady-state coordination and particularly perturbed coordination than originally reported. This was indeed the case. As can be appreciated from Figure 1, the half-cycle normalization procedure reduced artifacts in the form of undulations in the relative phase due to gradual and—especially relevant to the present case—sudden changes in the center of oscillation of the limb movements. As anticipated, this refined analysis smoothened the post-perturbation fit procedure and led to reduced data exclusion (only 8% as opposed to 21% in the original study [21]) and yielded estimates of the relaxation parameter λ that were more sensitive to the amplitude manipulations: Now congruent significant main effects of amplitude for λ and SDφ were obtained, both in the expected direction given coupled-oscillator dynamics [2] and previous studies [22,23,24,43], whereas in [21] this was only the case for SDφ. Clearly, coordinative stability was lower for smaller movement amplitudes, which complements previous findings that failed to observe amplitude effects in characteristics of transient, non-stationary bimanual performance [16,19,21], or multifrequency tapping [20]. While the mediating role of movement amplitude in for instance pattern transitions [6] remains questionable [19], the present steady-state and perturbation data clearly show amplitude dependency: Moving with larger amplitudes is associated with stronger coupling [22,23] and hence more stable coordination [21,22,23,24,43]. 

Before we proceed, it is important to recall that these findings were based on the data of six participants only. Although interpretations are thus to be made with caution, they do provide strong initial incentives for rekindling empirical research on the role of amplitude in coordinative stability. For instance, the present experiment may be rerun with a larger sample of participants and some further refinements. Regarding the latter, recall for instance that to manipulate the amplitude, endpoint targets were provided (see Section 2.2), which also implies (1) a constrained location of oscillation in space, and (2) more or less aiming behavior, involving anchoring of movement-reversal locations [11]. In the next experiment, amplitude manipulations without endpoint targets (as in [22,23]) are to be preferred.

In terms of practical implications, such findings can potentially be capitalized upon when seeking to improve interlimb or between-person coordination, as increasing the movement amplitude(s) is quite easy to achieve in for instance training sessions. An example may be found in the rehabilitation of arm function after stroke, in the context of which bilateral arm training is seen by some as a viable alternative to constrained induced movement therapy, a form of unilateral training [44,45,46]. By increasing the amplitude with which patients perform rhythmical bimanual exercises, it might be possible to increase the interlimb coupling strength [47] and thereby the stability of performance, which may accelerate upper-arm function recovery. In this context it would be particularly interesting to investigate under which amplitude conditions the most affected arm benefits most from the less affected arm, if at all. 

The ubiquitous finding that steady-state antiphase coordination is less stable (as indicated by higher SDφ) than in-phase coordination was corroborated, with SDφ values being qualitatively and quantitatively comparable to those in Post et al. [21]. However, the stability of antiphase and in-phase coordination patterns did not differ significantly in terms of the stability parameter λ (similar to [21,22,37,40]). It might be that within the present empirical design λ was not sufficiently sensitive to capture such differences (see also [22,48]) or that the stability of interlimb coordination after perturbation did not differ much between both patterns. With regard to the latter, it is important to realize that participants may particularly tend towards recapturing the coordination pattern around the left and right arms’ common movement-reversal point (‘anchor points’ [49]). Since in the experiment of Post et al. [21] the movements in both patterns were paced by a metronome, this might have invoked a locking of the common movement reversal to a ‘local’ beep (a form of common auditory-motor coordination governing interlimb coordination), thereby yielding similar relaxation processes for the in-phase and antiphase interlimb coordination (see also [21]). 

Given the observed lack of difference in stability (as depicted by λ) between in-phase and antiphase, it is also important to stipulate that the current task implies a mechanical coupling between the limbs. Specifically, while performing horizontal lower arm oscillations in an antiphase pattern (i.e., the arms move in the same direction in space), especially with larger (forceful) movements reaction forces imply the trunk to oscillate in the direction opposite of the arms [50]. Given the more stringent inertial properties of the trunk compared to the lower arms, these ‘extra’ trunk oscillations may work in a mechanically stabilizing way [50]. Note that such mechanical coupling does not—or at least to a much lesser extent—apply in case of more distal movements (e.g., rods, wrists, fingers). Certainly in view of potential applications (see above), it would therefore be useful to examine possible stabilizing benefits of mechanical coupling related to inertial properties of limbs and other components. 

When comparing our steady-state prescribed amplitude stability findings depicted in Figure 3a to those in the analogous Figure 3 of Post et al. [21], the obtained SDφ values were generally comparable, except for the higher frequencies (>1.25 Hz) for in-phase coordination, where the refined analysis yielded much lower values (indicating higher stability). Hence, whereas for antiphase coordination the commonly observed increase in SDφ with frequency was corroborated (see closed circles in Figure 3a), the reanalysis revealed that SDφ of the in-phase pattern significantly decreased with increasing frequency (see open circles in Figure 3a). This is a remarkable finding, as the vast majority of previous studies reported a (non-)significant increase of the variability of in-phase coordination with increasing movement frequency (as could also be expected from coupled-oscillator models [2]). Interestingly, a recent on-water crew rowing study also revealed the enhanced stability of steady-state in-phase between-rower coordination at a higher movement frequency [51], a task in which the amplitudes (i.e., oar excursions) are constrained as well. It is crucial here to realize that the stabilizing effect of frequency on in-phase coordinative stability was only present for the prescribed amplitude conditions (Figure 3a) and not for the ‘free’ amplitude condition (Figure 3b). As with increasing movement frequency self-chosen amplitudes typically drop (see Introduction), pattern stability is indeed likely to decrease. This was evident in the present data: For both in-phase and antiphase patterns ‘free’ movement amplitudes dropped with increasing movement frequency (see open circles in Figure 2), accompanied by rather similar and diminishing steady-state coordinative stability for in-phase and antiphase coordination, respectively (see Figure 3b). With prescribed amplitudes the coordinative stability of antiphase coordination indeed diminished with increasing frequency, suggesting a genuine stability dropping effect of higher movement frequency [2,6] as the reduced coordinative stability with increasing frequencies could not be mediated by a gradual change in amplitude. Unexpectedly, though, for prescribed amplitudes in in-phase coordination, a completely different pattern of results was observed: While at higher movement frequency the amplitude could not drop, the steady-state in-phase coordination further stabilized rather than maintained or diminished stability (Figure 3a), which contradicts both ‘frequency-mediated’ and ‘amplitude-mediated’ explanations [2,7,16,17,18,19,20,21].

At first glance, the in-phase stabilization with frequency may seem paradoxical, but may tentatively be explained (at least partly) in terms of the frequency-amplitude trade-off itself: Higher frequencies naturally would invoke smaller movement amplitudes, but when amplitude remains the same, the stabilizing effect of the relatively larger amplitude (for a certain frequency) induces stronger interlimb coupling and hence stabilizes coordination. A second tentative account, and one with greater implications, may be that the coupling strength (and hence pattern stability) may depend more strongly on something else than the mere frequency and amplitude of the oscillating components. In terms of coupled-oscillator modeling [2,7], this could for instance imply that frequency-related adaptations of (1) non-linear damping of the oscillatory behavior (e.g., [52]) and/or (2) coupling parameters come into play. In this respect it is interesting to mention that mathematically it has been shown that for in-phase coordination process 1 is a main mediator and for antiphase it is actually both 1 and 2 (e.g., [53]), which would indeed be in line with the currently obtained difference between in-phase and antiphase in terms of their frequency-related dynamics. Importantly, this explanation would entail that coupling parameters actually change as a function of time and behavior [3], whereas in the vast majority of pertinent studies the coupling parameters are assumed to be constant. The possibility of having time-varying coupling parameters constitutes a challenge for both further theoretical and empirical analysis.

The results of our reanalysis demonstrate that half-cycle normalization can be of critical importance in both transient and steady-state analyses of oscillatory signals. The extent to which this is the case depends on the degree to which movement oscillation centers and amplitudes change within and between trials. This was clearly an issue in the present dataset, but may be less of a problem in other (published) experimental data in which movements were performed more consistently around a steady center, such as in studies involving pendulum swinging in the gravitational plane (e.g., [9]). At the same time, in these studies the role of amplitude has typically not (sufficiently) been considered, let alone checked for (see also Introduction). The results of our reanalysis illustrate that it is crucial to always closely inspect the trial data a priori, in this case to check for performed amplitudes and for potential changes in the oscillation center so as to carefully evaluate the need for half-cycle normalization (or another form of oscillation center normalization [36]) prior to phase-angle determination.

## 5. Conclusions

Our refined analysis of the data of Post et al. [21] revealed that smaller prescribed amplitudes yielded lower coordinative stability. In addition, the observed frequency effects were not a mere consequence of the frequency manipulation as such but could be largely ascribed to the performed amplitude. These findings breathe new life into the claim that the stability of interlimb coordination depends more on (frequency-induced alterations of) movement amplitude than on movement frequency per se [22,23,43]. Furthermore, for prescribed small amplitudes in-phase coordination in fact stabilized (rather than mitigated) with higher movement frequencies. Although at this stage the results of the reanalysis of the data of Post et al. [21] and corresponding interpretations should be treated with caution (in view of the small number of participants), they do provide strong grounds for rekindling empirical research on the role of amplitude in the stability of rhythmical interlimb coordination. In this context, previous data other than those of Post et al. [21] may also be usefully subjected to refined analyses, since there are more previous studies in which certain effects may have been obscured and/or confounded due to (1) unnoticed changes in movement amplitude, and/or (2) artifacts related to inappropriate data normalization.

## Figures and Tables

**Figure 1 brainsci-10-00724-f001:**
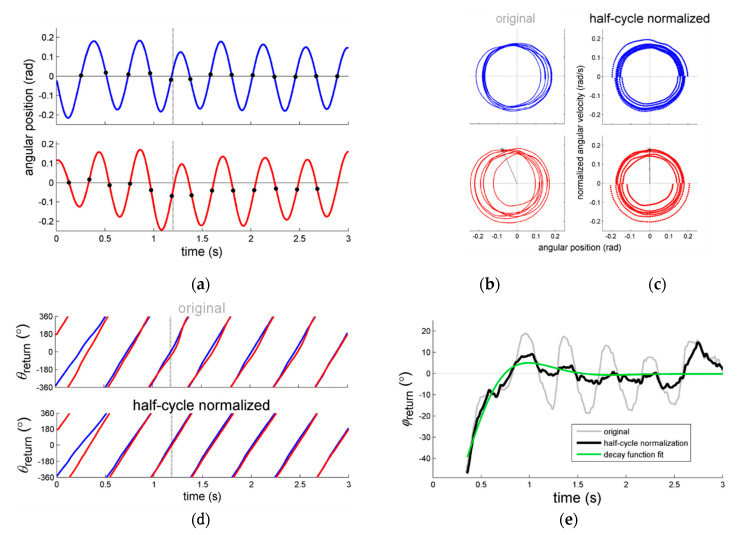
(**a**–**e**). A typical example of data directly after perturbation release: (**a**) Angular position data of both arms (blue = left arm, red = right arm) and the corresponding phase portraits as (**b**) determined in the original paper, and (**c**) after half-cycle normalization was applied; straight lines exemplify the calculated phase angle at *t* = 1.2 s; (**d**) depicts the corresponding phase angles $\theta_{\mathrm{return}}$ for the original data (upper panel) and after half-cycle normalization (lower panel); and (**e**) shows the resultant relative-phase signal $\varphi_{\mathrm{return}}$ as determined by Post et al. [21] (grey line) and after half-cycle normalization (black line). Green line = fitted decay function. The trial condition was in-phase, the frequency = 2.25 Hz, and the prescribed amplitude = 0.2 rad. See main text for further explanation.

**Figure 2 brainsci-10-00724-f002:**
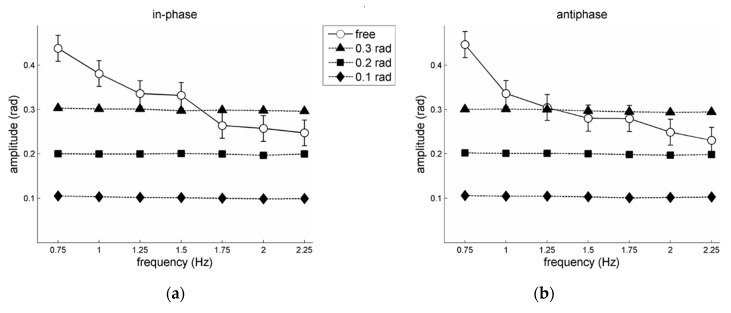
Performed movement amplitudes averaged over participants, as a function of frequency and amplitude conditions, for (**a**) in-phase and (**b**) antiphase coordination; free = unprescribed amplitude condition. Error bars depict between-subjects standard errors.

**Figure 3 brainsci-10-00724-f003:**
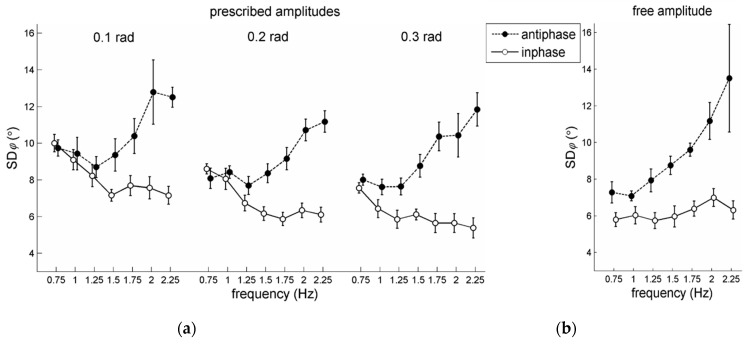
Stability of steady-state coordination (SDφ, with lower values indicating superior stability) for in-phase and antiphase coordination as a function of frequency conditions, for (**a**) prescribed amplitude and (**b**) free amplitude conditions. Error bars depict between-subjects standard errors.

**Figure 4 brainsci-10-00724-f004:**
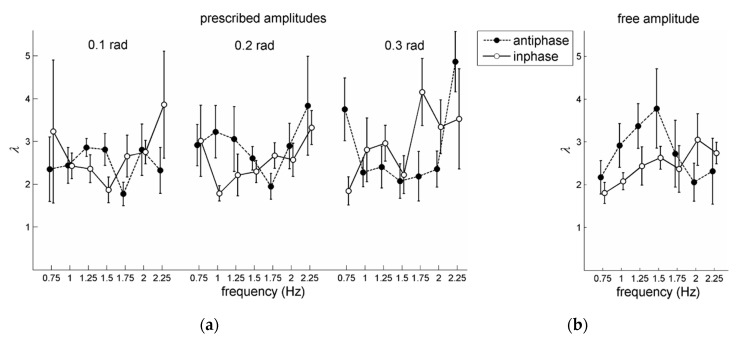
Post-perturbation coordinative stability (λ, with higher values indicating superior stability) for in-phase and antiphase coordination as a function of frequency conditions, for (**a**) prescribed amplitude and (**b**) free amplitude conditions. Error bars depict between-subjects standard errors.

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
