# Peer review of "A Re-Appraisal of the Effect of Amplitude on the Stability of Interlimb Coordination Based on Tightened Normalization Procedures"

_brainsci, 2020, doi:10.3390/brainsci10100724_

Round 1
Reviewer 1 Report
Overall the paper is structured well and clear. It would be good to see how these potential changes in the reanalysis have a practical implications for moving forward with this research field. In addition the critique of the approaches taken to reanalysis this data could be clear provided a sound rationale for this approach.
Abstract - A greater rationale is required in the abstract as to why the work of Post et al (2000) needed to be re-analysed and what was this assumption based upon. In addition a clear rationale for the new approach would be beneficial.
Introduction - The introduction is formatted well and clearly overviews the development that has led to the this body of work. The rationale for the new analysis is clear but it would be good to see greater analysis and critique of the current body of work to support your approach.
Methods - Line 134 the word 'was' should be 'were'. In addition statistical analysis section needs to be explained further
Results - Figure 2, there is inconsistency with the axis labels between A & B. The descriptions presented are clear but there are again inconsistency with how the significant data is described (especially p values).
Discussion - The discussion explores the findings of the study well links to the previous work of Post et al (2000). It would be good to see how these findings have a practical implication and future directions within this field would benefit the discussion.
Author Response
Overall the paper is structured well and clear. It would be good to see how these potential changes in the reanalysis have a practical implications for moving forward with this research field.
RESPONSE: Thank you for your useful suggestion, which we have followed up in the Discussion section by adding an example of a possible application in the rehabilitation of upper-extremity function after stroke using bilateral arm training. However, in view of the small number of participants (see also the comments of reviewer 2), there is a need to treat the findings of the reanalysis with caution and to avoid presenting them as conclusive. Hence, with exerting an appropriate degree of caution with regard to our findings, we have decided to keep the practical implications part concise and focused on the concepts underlying these implications. It reads as follows: “In terms of practical implications, such findings can potentially be capitalized upon when seeking to improve interlimb or between-person coordination, as increasing the movement amplitude(s) is quite easy to achieve in for instance training sessions. An example may be found in the rehabilitation of arm function after stroke, in the context of which bilateral arm training is seen as a viable alternative to constrained induced movement therapy, a form of unilateral training [Lee, et al., 2017; Van Delden et al. , 2012; McCombe et al. 2008] By increasing the amplitude with which patients perform rhythmical bimanual exercises, it might be possible to increase the interlimb coupling strength [ van Delden, et al., 2015] and thereby the stability of performance, which may accelerate upper-arm function recovery. In this context it would be particularly interesting to investigate under which amplitude conditions the most affected arm benefits most from the less affected arm, if at all.” (Discussion, 3rd paragraph)
In addition the critique of the approaches taken to reanalysis this data could be clear provided a sound rationale for this approach. Abstract - A greater rationale is required in the abstract as to why the work of Post et al (2000) needed to be re-analysed and what was this assumption based upon. In addition a clear rationale for the new approach would be beneficial.
RESPONSE: This is a valid point that was also raised by reviewer 4. Throughout the revised manuscript, including the abstract, we have strengthened both the rationale for re-analyzing the work of Post et al. and the rationale for the new approach (e.g., by rewriting sections 1.1. ‘Incentives for…’ and parts of 1.2 ‘Normalization issues’).
Introduction - The introduction is formatted well and clearly overviews the development that has led to the this body of work. The rationale for the new analysis is clear but it would be good to see greater analysis and critique of the current body of work to support your approach.
RESPONSE: Just a limited number of studies have been published to date examining the role of amplitude in relation to coordinative stability. To point the reader to the relevant literature on this topic, we have added the following sentence at the end of the Introduction: “… previous studies have indeed found indications that moving with larger amplitudes relates to enhanced interlimb [Post et al 2000; De Poel et al 2008; Peper et al 2008 2009; Ryu et Buchanan 2004; Buchanan & Ryu 2012; Spijkers &Heuer 1995 ] and perceptuo-motor coupling [De Boer et al. 2103; Varlet et al 2012; Kudo et al 2006]”
Methods - Line 134 the word 'was' should be 'were'. In addition statistical analysis section needs to be explained further
RESPONSE: We changed ‘was’ into ‘were’; thank you for spotting this mistake. In addition, we added a heading ‘statistical analysis’. As specific choices regarding post hoc analysis depended on the interaction effects obtained in the omnibus ANOVA, we chose to provide these details at the spot where results are presented, which in our view is clearer for the reader. In the ‘statistical analysis’ section we specifically indicate this as follows: ‘More specifics for each test follow in the Results section.’
Results - Figure 2, there is inconsistency with the axis labels between A & B. The descriptions presented are clear but there are again inconsistency with how the significant data is described (especially p values).
RESPONSE: In the revised version we changed the appearance of Figures 2 and 3, which was done in accordance with the suggestion of reviewer 2 to add error bars. In our view this new figure structure also made it more accessible for the reader. We have now also included outcomes of the statistical tests on performed amplitudes in the free amplitude condition to validate the original admittedly qualitative description of the results of Figure 2 with solid statistical evidence.
Discussion - The discussion explores the findings of the study well links to the previous work of Post et al (2000). It would be good to see how these findings have a practical implication and future directions within this field would benefit the discussion.
RESPONSE: Done, as specified in our response to the first point above.
Reviewer 2 Report
Thank you for giving me the opportunity to review this article. In this study, the authors re-analyzed a dataset published by Post et al. (2000). In the original analyses, Post et al. (2000) found that coordination patterns were more variable for smaller prescribed amplitudes, but amplitude did not affect stability. The authors wanted to re-analyses these data because there is recent evidence that interlimb coupling strength actually scales with movement amplitude. With a different analyses procedure better controlling for artifacts, they found that amplitude played a role in movement variability and relaxation (i.e., time to return to the initial synchronization after a perturbation). More specifically, stability increased with amplitude. Also, performance became less stable with increasing frequency when in anti-phase, and more stable with increasing frequencies when in-phase. The article is clear and well written. The study is very interesting and this new analysis of published data sheds light on important methodological and theoretical issues in human movement science.
Although I like the fact that the authors reanalyzed existing data rather than conducted a new study, I am concerned by the small number of participants. When the original study was published, reproducibility and statistical power tended to be overlooked. I am not convinced that Anovas with only six participants might be interpretable. The authors should explain why they have so few participants and to what extent they think that their results are valid. Results can be driven by the performance of one or two participants. The authors should provide some information about interindividual variability. At the very least, error bars should be added to the graphs, or the individual results might be added in supplementary material.
I suggest that the authors revise the article taking a new twist. Currently it is presented as an experimental article, but it would probably be justified to present it as a methodological article. This would justify the small number of participants. Conclusions and interpretation of the results should be downplayed, as the study is clearly underpowered. This would not necessitate a complete rewriting of the manuscript as I find it clear and well written. However, the introduction and conclusion should state clearly that given the small number of participants, cautious interpretation is required. This way, the article would focus more on the different outcomes obtained with two different pre-processing methods, and less on the theoretical implications of the new findings. Further studies will confirm or infirm these results.
Other comments:
- Introduction
Although I find the first paragraph clear, I suggest that the authors start with a more general statement about rhythmical interlimb behavior. The first sentence is hard to grasp and naïve readers may not understand what they mean by “limited number of stable coordination patterns”.
Some sentences are too long (lines 46-49, 50-53) and could easily broken down.
“Why may this be problematic? Well, it is generally known etc.” (line 56): I find this conversational style awkward.
“increases (decreases)” (line 57-58): this wording is not easy to read. “Increase/decreases” would read better, or consider changing the formulation.
“both with (0.1, 0.2, and 0.3 rad) and without prescribed amplitude”: this sentence is hard to understand; it’s not obvious that the radians refer to the prescribed amplitude here.
“depend more on of amplitude” (line 125): there is an extra word
- Materials and Methods
More information about the participants would be welcome. Were they musicians, trained athletes etc.? This might be considered in relation with my point about the number of participants and inter-individual variability.
Why did the authors use the SD and not the coefficient of variation? I think it would be more appropriate as the mean of the relative phase might differ. Are the results the same with the CV?
The authors say that they no not consider the trials with a freely chosen frequency, but they are still presented in the results section. Please explain.
Line 199: I would add a sentence saying that ? will be used as an index of stability, and refer consistently to stability throughout the text
- Results
There is an inconsistency in the presentation of figure 3. The caption says that the relative phase value (?) is indicated on the y axis, but the axis title indicates the phase (?)
- Discussion
“a.k.a” (line 299): this should be spelled out
Author Response
Thank you for giving me the opportunity to review this article. In this study, the authors re-analyzed a dataset published by Post et al. (2000). In the original analyses, Post et al. (2000) found that coordination patterns were more variable for smaller prescribed amplitudes, but amplitude did not affect stability. The authors wanted to re-analyses these data because there is recent evidence that interlimb coupling strength actually scales with movement amplitude. With a different analyses procedure better controlling for artifacts, they found that amplitude played a role in movement variability and relaxation (i.e., time to return to the initial synchronization after a perturbation). More specifically, stability increased with amplitude. Also, performance became less stable with increasing frequency when in anti-phase, and more stable with increasing frequencies when in-phase. The article is clear and well written. The study is very interesting and this new analysis of published data sheds light on important methodological and theoretical issues in human movement science.
Although I like the fact that the authors reanalyzed existing data rather than conducted a new study, I am concerned by the small number of participants. When the original study was published, reproducibility and statistical power tended to be overlooked. I am not convinced that Anovas with only six participants might be interpretable. The authors should explain why they have so few participants and to what extent they think that their results are valid.
RESPONSE: First of all, thank you for your insightful comments. As regards your first point, the main reason at the time was that not many participants appeared to be able to perform all the experimental conditions to a sufficient degree (as was mentioned in section 2.1). In the revised version we have now provided some further explication in this regard. By the way, this is also the main reason underlying why the selected prescribed amplitudes were quite small: otherwise the participants could not perform the task at the highest frequencies. Although six participants is sufficient for proper ANOVA testing considering the repeated-measures design, we agree that the limited number of participants is a concern that needs to be taken into account when interpreting the results (see also the reply to reviewer 1 on this issue, and the reply to your final main comment). We have therefore exerted an appropriate degree of caution with regard to the interpretation and implications of our findings throughout the revised manuscript.
Results can be driven by the performance of one or two participants. The authors should provide some information about interindividual variability. At the very least, error bars should be added to the graphs, or the individual results might be added in supplementary material.
RESPONSE: Point well taken. In the revised version we have added error bars to the figures as requested. Moreover, we made individual participant data available in Tables as supplementary material (i.e., performed amplitudes and stability measures across repeated-measures conditions). As can be appreciated from these data, although a fair degree of between-participant variation was of course present, the patterning of results over the various repeated-measures conditions was across the board in fact quite consistent among the six participants.
I suggest that the authors revise the article taking a new twist. Currently it is presented as an experimental article, but it would probably be justified to present it as a methodological article. This would justify the small number of participants. Conclusions and interpretation of the results should be downplayed, as the study is clearly underpowered. This would not necessitate a complete rewriting of the manuscript as I find it clear and well written. However, the introduction and conclusion should state clearly that given the small number of participants, cautious interpretation is required. This way, the article would focus more on the different outcomes obtained with two different pre-processing methods, and less on the theoretical implications of the new findings. Further studies will confirm or infirm these results.
RESPONSE: We thank the reviewer for this useful suggestion. Although we have refrained from recasting our study from an experimental article into a methodological article, we have made several changes in the manuscript that collectively place more emphasis on the methods used and less emphasis on the results obtained. In the Introduction and Conclusion sections of the revised version we now clearly acknowledge that the reanalyzed data involved a small number of participants and that therefore the results of the reanalysis should be treated and interpreted with caution. Instead of presenting the results of the reanalysis as firm conclusions, we know emphasize that they provide a compelling reason to rekindle empirical research on the role of amplitude in the stability of rhythmical interlimb coordination: “Although at this stage the results of the reanalysis of the data of Post et al. [21] and corresponding interpretations should be treated with caution (in view of the small number of participants), they do provide strong grounds for rekindling empirical research on the role of amplitude in the stability of rhythmical interlimb coordination.”
Other comments:
- Introduction
Although I find the first paragraph clear, I suggest that the authors start with a more general statement about rhythmical interlimb behavior. The first sentence is hard to grasp and naïve readers may not understand what they mean by “limited number of stable coordination patterns”.
RESPONSE: We followed the suggestion and provided a more general start of the introduction. Much better indeed, thanks!
Some sentences are too long (lines 46-49, 50-53) and could easily broken down.
RESPONSE: Known issue. We made an attempt to shorten long sentences throughout the revised manuscript. This improved its readability.
“Why may this be problematic? Well, it is generally known etc.” (line 56): I find this conversational style awkward.
RESPONSE: We have reworked sentences containing conversational style throughout the revision.
“increases (decreases)” (line 57-58): this wording is not easy to read. “Increase/decreases” would read better, or consider changing the formulation.
RESPONSE: To avoid such potential confusion we rephrased that sentence as follows: “…. it is generally known that higher frequencies are accompanied by smaller characteristic amplitudes and vice versa that larger amplitudes are accompanied by lower frequencies.”
“both with (0.1, 0.2, and 0.3 rad) and without prescribed amplitude”: this sentence is hard to understand; it’s not obvious that the radians refer to the prescribed amplitude here.
RESPONSE: We rephrased this sentence as follows: “….at three prescribed amplitude conditions (i.e., prescribed amplitudes were 0.1, 0.2, and 0.3 rad) and a ‘free’ unprescribed amplitude condition.”
“depend more on of amplitude” (line 125): there is an extra word
RESPONSE: Thanks for spotting this typo, we have deleted ‘of’.
- Materials and Methods
More information about the participants would be welcome. Were they musicians, trained athletes etc.? This might be considered in relation with my point about the number of participants and inter-individual variability.
RESPONSE: In line with this point (and see also related previous points), in the revised version we added that the participants were selected based on whether they could perform the tasks at all, as is now explicitly stated in section 2.1: ‘… six participants were selected from a larger pool based solely on their ability to perform in-phase and antiphase movements in the experimental amplitude and frequency ranges as determined from a preceding test session (see [21]).’ Hence, at the time of the experiment there were also some candidate participants that could not properly perform the task in all conditions.
Why did the authors use the SD and not the coefficient of variation? I think it would be more appropriate as the mean of the relative phase might differ. Are the results the same with the CV?
RESPONSE: Not only is the circular standard deviation of relative-phase time series a key common outcome measure of steady-state pattern stability, reported in hundreds of papers, including the CV of relative-phase time series and compare that across in-phase and antiphase coordination conditions would also be problematic as the circular mean relative phase differs by approximately 180 degrees, hindering a fair comparison. Moreover, CV entails a division by the mean circular relative phase, which implies problematic division by values of or around zero for in-phase coordination. Results will likely be different, and non-interpretable from a steady-state coordinative stability perspective.
The authors say that they no not consider the trials with a freely chosen frequency, but they are still presented in the results section. Please explain.
RESPONSE: We are afraid this was a misread: Results regarding free frequency conditions are not presented; those regarding the free amplitude conditions were.
Line 199: I would add a sentence saying that ? will be used as an index of stability, and refer consistently to stability throughout the text
RESPONSE: We followed this suggestion and in the remainder of the text clarified ? where applicable. Note that SD ? was also adopted as a stability measure, namely for steady-state performance (as was also done in the original paper of Post et al, 2000)
- Results
There is an inconsistency in the presentation of figure 3. The caption says that the relative phase value (?) is indicated on the y axis, but the axis title indicates the phase (?)
RESPONSE: We now see the Ï•-symbol (phi, representing relative-phase evolution) in the axis label (which is different from ? symbol, theta, representing phase evolution) deviates from the ?-symbol in the main text, which probably caused this confusion. In the revised version we used consistent phi-symbols in text and figures.
- Discussion
“a.k.a” (line 299): this should be spelled out
RESPONSE: We decided to remove ‘a.k.a.’ altogether, without loss of meaning.
Reviewer 3 Report
The article addresses the topic of stability of interlimb coordination.
Material and methods. It is not clear if there is a consent to use the data reported by Post et al. (Relative phase dynamics in perturbated limb coordination: The effects of frequency and amplitude.
Conclusion. What are the practical implications of this study? Are there some consequences for clinical use?
Some sentences should be rephrased (for example, Well, it is generally known...-line 55).
Review line 91, including the data presented in reference no 28.
Line 131. Some data included in reference no 21 should be presented.
Author Response
The article addresses the topic of stability of interlimb coordination.
Material and methods. It is not clear if there is a consent to use the data reported by Post et al. (Relative phase dynamics in perturbated limb coordination: The effects of frequency and amplitude.
RESPONSE: As stated in section 2.1, at the time of the experiment the participants provided informed consent according to the declaration of Helsinki. Note that the goal of the current study did not differ from that of the original study (i.e., the goal which the participants signed consent for), which justifies the (re-)use of the data.
Conclusion. What are the practical implications of this study? Are there some consequences for clinical use?
RESPONSE: Yes, there might be implications for clinical practice, as specified in the Discussion section by adding an example of a possible application in the rehabilitation of upper-extremity function after stroke using bilateral arm training. However, in view of the small number of participants, there is a need to treat the findings of the reanalysis with caution and to avoid presenting them as conclusive. Hence, with exerting an appropriate degree of caution with regard to our findings, we have decided to keep the practical implications part concise and focused on the concepts underlying these implications. It reads as follows: “In terms of practical implications, such findings can potentially be capitalized upon when seeking to improve interlimb or between-person coordination, as increasing the movement amplitude(s) is quite easy to achieve in for instance training sessions. An example may be found in the rehabilitation of arm function after stroke, in the context of which bilateral arm training is seen as a viable alternative to constrained induced movement therapy, a form of unilateral training [Lee, et al., 2017; Van Delden et al. , 2012; McCombe et al. 2008] By increasing the amplitude with which patients perform rhythmical bimanual exercises, it might be possible to increase the interlimb coupling strength [ van Delden, et al., 2015] and thereby the stability of performance, which may accelerate upper-arm function recovery. In this context it would be particularly interesting to investigate under which amplitude conditions the most affected arm benefits most from the less affected arm, if at all.” (Discussion, 3rd paragraph)
Some sentences should be rephrased (for example, Well, it is generally known...-line 55).
RESPONSE: We have reworked sentences containing conversational style throughout the revision.
Review line 91, including the data presented in reference no 28.
RESPONSE: Unfortunately this comment was not clear to us. Perhaps a typo was made regarding reference number?
Line 131. Some data included in reference no 21 should be presented.
RESPONSE: Also this comment is unfortunately somewhat unclear to us. Exemplary data is provided in Figure 1. Or does it perhaps refer to making data files available?
Reviewer 4 Report
Manuscript Review:
Brain Sciences
Stability of Interlimb Coordination: A Re-appraisal of the Role of Amplitude Based on Refined Analyses
The authors re-analyzed some former data controlling for drift in the center of amplitude of an oscillatory movement in order to better examine changes in amplitude that result from changes in frequency. The previous analysis did not show a strong influence of amplitude on stability. With the new, normalized data, there was a significant effect of both frequency and amplitude, suggesting that both play a role in the stabilization of multi-limb coordination.
I have no major issues or comments for this manuscript. I think it is one of the most well-written manuscripts I have ever reviewed.
Minor comments:
Title: “…based on refined analyses.” I wonder if this could be more specific. Maybe something about “half-cycle normalization”? Just to give the reader (and searcher) a bit more about the main idea of the paper?
Lines 65-66: Where does the Post et al paper fit in with the literature review. It is only first mentioned on line 65 as a statement that the data here will be re-analyzed. Why this paper. The reader is left wondering why not re-analyze the data from sources 16, 19 and 20, previously mentioned as papers looking at the role of amplitude. It might be good to introduce the Post paper prior to the paragraph on its re-analysis.
Lines 74-76: It is not clear here what “limited the perturbation outcomes” means.
Line 123: I found the statement starting with “To anticipate, our findings ….” A bit odd to read. It’s maybe not so common to be given a spoiler of the main findings here?
Line 125: I think there is an extra “of” before amplitude here.
Author Response
The authors re-analyzed some former data controlling for drift in the center of amplitude of an oscillatory movement in order to better examine changes in amplitude that result from changes in frequency. The previous analysis did not show a strong influence of amplitude on stability. With the new, normalized data, there was a significant effect of both frequency and amplitude, suggesting that both play a role in the stabilization of multi-limb coordination.
I have no major issues or comments for this manuscript. I think it is one of the most well-written manuscripts I have ever reviewed.
RESPONSE: Thanks for these kind comments. We actually feel sorry for you if our manuscript was one of the most well-written manuscripts you have ever reviewed! Keep up the good work!
Minor comments:
Title: “…based on refined analyses.” I wonder if this could be more specific. Maybe something about “half-cycle normalization”? Just to give the reader (and searcher) a bit more about the main idea of the paper?
RESPONSE: Excellent suggestion, we have rephrased the title as follows: “A Re-appraisal of the Effect of Amplitude on the Stability of Interlimb Coordination Based on Tightened Normalization Procedures”
Lines 65-66: Where does the Post et al paper fit in with the literature review. It is only first mentioned on line 65 as a statement that the data here will be re-analyzed. Why this paper. The reader is left wondering why not re-analyze the data from sources 16, 19 and 20, previously mentioned as papers looking at the role of amplitude. It might be good to introduce the Post paper prior to the paragraph on its re-analysis.
RESPONSE: This is a valid point that was also raised by reviewer 1. Throughout the revised manuscript, we have strengthened both the rationale for re-analyzing the work of Post et al. and the rationale for the new approach (e.g., by rewriting sections 1.1. ‘Incentives for…’ and parts of 1.2 ‘Normalization issues’).
Lines 74-76: It is not clear here what “limited the perturbation outcomes” means.
RESPONSE: No longer relevant as this part if the introduction was completely rewritten in light of your previous point.
Line 123: I found the statement starting with “To anticipate, our findings ….” A bit odd to read. It’s maybe not so common to be given a spoiler of the main findings here?
RESPONSE: Indeed not so common, but including the main findings actually adhered explicitly to the journal format requirements (see the paper template provided at the instructions for authors: https://www.mdpi.com/journal/brainsci/instructions)
Line 125: I think there is an extra “of” before amplitude here.
RESPONSE: Thanks for spotting this typo, we have deleted ‘of’.
Round 2
Reviewer 2 Report
Thank you for your clear responses.